# Enhancing Software Comments Readability Using Flesch Reading Ease Score

**Derar Eleyan** [1,*] , **Abed Othman** [2] and **Amna Eleyan** [3]

1 Applied Computing Department, Palestine Technical University Kadoorie, Tulkarem P.O. Box 7, Palestine
2 Computer Science Department, Birzeit University, Birzeit P.O. Box 14, Palestine; aothman@birzeit.edu
3 School of Computing and Mathematics, Manchester Metropolitan University, Manchester M15 6BH, UK;
  A.Eleyan@mmu.ac.uk
* Correspondence: d.eleyan@ptuk.edu.ps

**Abstract:** Comments are used to explain the meaning of code and ease communications between programmers themselves, quality assurance auditors, and code reviewers. A tool has been developed to help programmers write readable comments and measure their readability level. It is used to enhance software readability by providing alternatives to both keywords and comment statements from a local database and an online dictionary. It is also a word-finding query engine for developers. Readability level is measured using three different formulas: the fog index, the Flesch reading ease score, and Flesch–Kincaid grade levels. A questionnaire has been distributed to 42 programmers and 35 students to compare the readability aspect between both new comments written by the tool and the original comments written by previous programmers and developers. Programmers stated that the comments from the proposed tool had fewer complex words and took less time to read and understand. Nevertheless, this did not significantly affect the understandability of the text, as programmers normally have quite a high level of English. However, the results from students show that the tool affects the understandability of text and the time taken to read it, while text complexity results show that the tool makes new comment text that is more readable by changing the three studied variables.

**Keywords:** Flesch–Kincaid; Flesch reading ease score; fog index; readability; software code; software comment software quality

## 1. Introduction

Code writing, when developing a software life cycle, is considered an important phase. This is because the next phases depend on the current one, especially in the software maintenance phase, which needs to take code maintainability, readability, and reusability into consideration. Because of this, we should be careful with code readability, which is related to the three terms mentioned before. In addition, readability becomes a key factor in software quality [1]. On one hand, the readability of source code is grouped with its documentation to influence software maintainability. On the other hand, researchers noted that it takes a long time to understand the code after or while developing compared with other software maintenance activities [1,2] and that source code readability and documentation readability are both critical to the maintainability of a project. Other researchers have noted that the act of reading code is the most time-consuming component of all maintenance activities. Among the factors that affect this are comments, which are defined as embedded documents and useful artifacts related to code quality [2,3]. The existence of these comments is necessary for program comprehension, especially when the maintainers differ from the programmers or main developers. For example, comments were involved in about 15–20% of Mozilla source code [2].

This research is concerned with developing a tool for enhancing source code readability, using predefined formulas to amend the source code's quality by increasing the readability level of code comments. This tool will provide the programmer with suggestions to improve the readability level of written comments. This tool will be a "comments readability system" (CRS). This is a tool that can be used by programmers to perform two main actions—to verify the current comment readability levels and to provide suggestions of alternatives for readability improvements to make comments more understandable.

This tool will use three different formulas to check the current comment complexity level; by this immediate evaluation, new terms or words will be suggested to the programmer. The Flesch reading and Gunning fog formulas, with modified measuring parameters, alongside a database of the suggested complex words, will be used to assess the writing of comments. Then, the system will mark each word that does not satisfy the preconfigured criteria and score each comment statement. This will provide an opportunity for the programmer to go through each marked word and provide readable alternatives from the database or corpus, which will be directly connected to the tool. This tool was evaluated by several programmers and developers who we requested to use the tool and provide their own feedback; a results analysis is shown in the analysis section. The following sections will discuss the motivation behind this research and its contribution to the state of the art, the methodology conducted, the results, the discussion and the research conclusions.

## 2. Literature Review

### 2.1. Software Quality

Software quality (SQ) is an important factor in the software life cycle, affecting the cost, time and user requirements of software development. Developers always aspire and exert efforts to develop software of high quality, which is free of faults and errors [1,2,4–6].

The software quality definition, according to the International Organization for Standardization (ISO 9126) varies from matching system requirements to the capability of covering dynamic changes. Previously, software quality was used to measure defects in software and check the potential of systems to match user requirements, required functions, and the capability of systems to accept future requirements or changes [6–9].

SQ is normally assessed by user experience who is interested in external quality and software features and the programmer or developer who is interested in software development especially about the code [4,6,8–10].

The software development time is short if its compared with code maintenance and enhancements which consumed over 70% of the total life cycle and cost of a software product as shown in Figure 1 [5,6,11]. Code maintenance becomes more difficult without comments or text documentation about the program. The probability of software development bugs occurrence increases if there are no communications tools among programmers to explain their code. Therefore, the comments are an essential factor that help and support programmers to understand the code especially if the software is used in critical fields [1,11].

### 2.2. Code Review

A code review is considered an important stage, role function in modern software development [12] which is used to identify bugs or errors in source code and to improve code quality [13,14]. Software quality is the function of the degree of reviews made during the early life of a software development process which requires comments to easily understand the software and fasten the maintenance process [15].

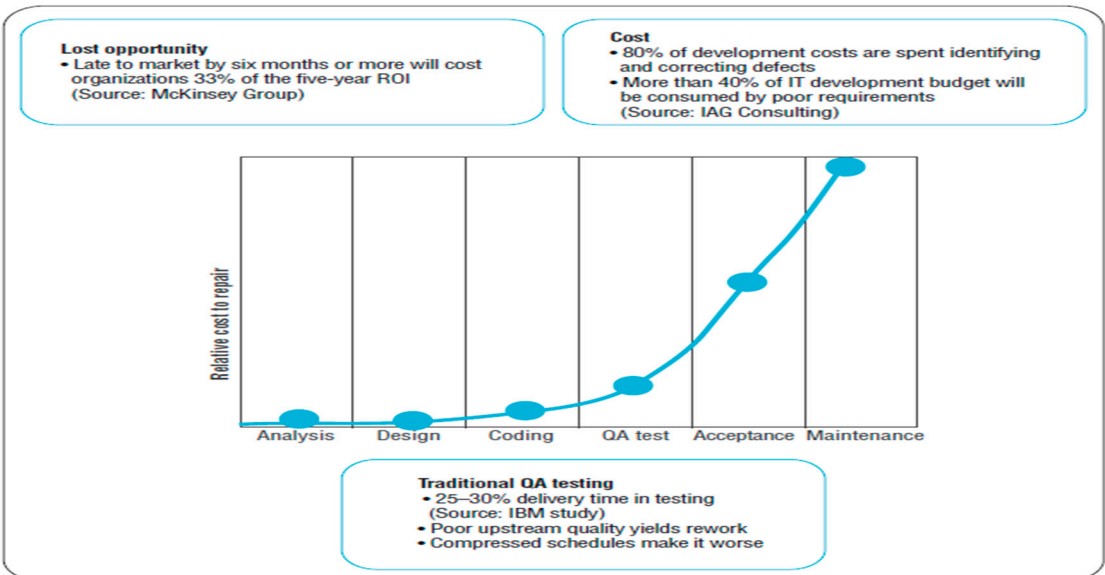

**Figure 1.** Opportunities to optimize quality and time to market exist throughout the application development lifecycle [6].

*2.3. Code Readability and Comments*

Software readability or "code readability" is an important part to assess software maintainability: "How quickly can a new developer understand the code?" [4,6,10,16–18]. It has various definitions; i.e., it is the human judgement of how easy a source code is understood by programmers, or the process by which a programmer makes sense of a programming code [8,16,19–22]. Most companies' developers are working as a team instead of working individually, thus most of the software codes are written by different teams; the code must be reusable and maintainable, so it is important that this code must be understandable. Therefore, readability becomes a priority, also it has always been the reason behind the maintainable code, so readability is needed. Thus, readability becomes a dominant topic in software engineering especially in software quality [3,11,23]. "Maintenance programmers" tasks are analogous to those of archaeologists who investigate some situations, trying to understand what they are looking at and how it all fits together. To do this, they must be careful to preserve the artifacts they find and respect and understand the cultural forces that produced them" [16]. The difficulty of understanding code influences the high cost of software maintenance [5]. Moreover, reducing the reusability of this source code especially a source code may be considered a readable one for a programmer but it might not be for another. This can be applied in two situations when others need to understand the code or the author of the code after a while (month, year), he/she could not remember the written code or took a long time to do. Most researches show that the reading the code and understanding it, takes a longer time compared to maintenance activities [16,21,22].

> *"Unfortunately, computer system documentation is difficult to understand, badly written, out-of-date or incomplete."* [24]

The complexity of the code and readability are not the same although they are closely interrelated; complexity is code property based on the problem domain, the method behind solving each scenario will not be avoided completely in developing the process. Whereas readability is, an accidental property that can be avoided independent of the problem domain or problem complexity [21]. Code readability is not easy to measure by a deterministic function similar to maintainability, reliability, and reusability. Readability measures formula care about and changed by independent individual elements, such as identifiers, statements, comments, indentation, and program style. However, complexity measurements depending on static components as well as dynamic interactions among program components [21,25,26].

Although most of the evidences show the importance of commenting on the code, some programmers adopt the opposite point of this view. They say the good code does not need to comment on it because the code itself is clear and understandable [8,27] says "The frequency of comments sometimes reflects the poor quality of code. When you feel compelled to add a comment, consider rewriting the code to make it clearer". Developers don't need to comment on every line of code, because when the programmer needs to comment on every line of code to make it understandable, this might indicate that the code is low quality or have lacked in structure, in this case, programmers should rewrite it instead of commented it. However, as mentioned before comments are necessary for the development process and maintenance and also to make the code more readable even when the code itself is low quality [4,11,19,21,28].

Students study codes from examples in textbooks, outline documents, other colleague's code; these resources and others are used as learning tools in academic environments. These codes or programs can be unhelpful if their readability is too high or if it is too low. That makes understanding the code hard and increases the difficulty of maintainability or reusability of these codes [25].

Software code is divided into two types: the code itself, and comments on the code. Comments are a block of source that the compiler ignores in the compilation process, so you can put in it any information that you desire. However, a programmer should keep in mind that the main aim of using these comments are notation of the source code and help the developer and other software implementer team members to understand the source code in the process of upgrading or bugs fixing or to make software fits with new or modified requirements. In other words, to aid in software maintenance and therefore reduce maintenance costs. Several researchers have conducted experiments showing that commented code is easier to understand than code without comments. Besides, these comments are used as internal documentation for code and system [19,29].

We should keep in mind that the comments are used to make the code more readable not to replace it. In addition, if the code is inadequate we should make changes to have an adequate code not to have a good comment. However, we can say that bad comments are worse than no comments at all, for this, "We should only be writing comments when they really add meaning" [29]. By the way, some concerns should be taken into consideration when writing comments as [10,29] mentions:

i    Quality of the comments is more important than quantity.
ii   Good comments explain why and how.
iii  The code should not be duplicated in the commenting.
iv   Comments should be clear and concise.
v    Functions should start with a block commenting
vi   Comments should be indented the same way as the rest of the code
vii  Comments should be used in a way that reads naturally before a function, not below.

There are different styles used in commenting, each language almost has its own commenting style. Most of which are set in green color, i.e., C language comments come in blocks between /* and */ and can span any number of lines. C++, C#, Java add the single line comment that follows //, and vb.net used single comma [29].

*2.4. Comment Types*

2.4.1. Documentary Comments

This type focuses on the history of the file system and main properties of this file like author, date, updated date, copyright, the introduction of class or source file.

### 2.4.2. Functional Comments

These types of comments are used to serve a single process in the development cycle. This famous example used here is "to do". In addition, there are three positions we can find it [18]:

i　　Bug description;
ii　　Notes to co-developers;
iii　　Performance/feature;
iv　　improvement possibilities.

### 2.4.3. Explanatory Comments

This type of comment is the most commonly used one. Its function comes tied with its name to give more details about the item commented. However, there are many items, which have this type of comment [27] as follows:

i　　Startup code;
ii　　Exit code;
iii　　Subroutines and functions;
iv　　Long or complicated loops;
v　　Weird logic;
vi　　Regular expressions.

### 2.5. The Case against Commenting

### 2.5.1. Programmer's Hubris

Many programmers see that nothing is hard to do, everything is possible and easy to understand and will be developed easily. But in reality, this is not 100% correct; many programmers can't remember their code after a year for example. "Truth is, most of the time it will take a lot of time and effort to understand the undocumented code, even if the code has not been obfuscated intentionally" [18].

### 2.5.2. Laziness

"All-time saved by not commenting during the coding process is made up more than twice at least by inserting comments and providing documentation after the coding process is finished" [27]. Because nowadays there is software used to generate documentation by the automatic collection of comments, the writing comment will save the time of writing documentation after software development is completed [18].

### 2.5.3. The Deadline Problem

The project deadline will not be the problem of omitting comments, because the time taken to write comments that the developer wishes to save while he is in the development phase will be less than that one needed to fix bug with an uncommented code after a while [18,30].

### 2.6. Readability Formulas

### 2.6.1. Flesch Reading Ease

This formula designed by Rudolph Flesch to measure the difficulty of text document context, and is used as an indicator for grading the difficulty of understanding reading content in English. This grading depends on several factors that affect the text content, such as word length, sentence length, word form, and syllables or letters. This formula produces scores determining the readability level of the text as shown in Table 1 below [17,21,26,31].

**Table 1.** Flesch reading ease score to assess the ease of readability in a document [31].

| Score | School Level | Notes |
|---|---|---|
| 100.0–90.0 | 5th Grade | Very easy to read. Easily understood by an average 11-year-old student. |
| 90.0–80.0 | 6th Grade | Easy to read. Conversational English for consumers. |
| 80.0–70.0 | 7th Grade | Fairly easy to read |
| 70.0–60.0 | 8th & 9th Grade | Plain English. Easily understood by 13- to 15-year-old students |
| 60.0–50.0 | 10th to 12th Grade | Fairly difficult to read. |
| 50.0–30.0 | College | Difficult to read |
| 30.0–10.0 | College Graduate | Very difficult to read. Best understood by university graduates |
| 10.0–0.0 | Professional | Extremely difficult to read. Best understood by university graduates |

The following is the algorithm to determine Flesch reading ease [17,21,31].

✓ Calculate the average number of words used per sentence.
✓ Calculate the average number of syllables per word.
✓ Multiply the average number of syllables per word multiplied by 84.6 and subtract it from the average number of words multiplied by 1.015.
✓ Subtract the result from 206.835.

Algorithm: $206.835 - (1.015 \times average\_words\_sentence) - (84.6 \times verage\_syllables\_word)$

### 2.6.2. Flesch Kincaid

This is another formula of text readability measurement designed by Rudolph Flesch to use the same core measures (word length and sentence length) as Flesch reading ease but it uses different weighting factors. The following is the algorithm to determine the Flesch–Kincaid grade level [31,32]. Table 2 shows the scores generated and the correspondent meaning.

**Table 2.** Formula of Flesch–Kincaid [4].

| Score | Note |
|---|---|
| 90–100 | Easily understood by an average 11-year-old student |
| 60–70 | Easily understood by 13- to 15-year-old students |
| 0–30 | Best understood by university graduates |

✓ Calculate the average number of words used per sentence.
✓ Calculate the average number of syllables per word.
✓ Multiply the average number of words by 0.39 and add it to the average number of syllables per word multiplied by 11.8.
✓ Subtract 15.50 from the result.
✓ Algorithm: $(0.39 \times average\_words\_sentence) + (11.8 \times average\_syllables\_word) - 15.9$

In the Flesch reading ease test, higher scores indicate the text is easier to read; lower numbers scores indicate the text is difficult to read. Scores can be interpreted as shown in Table 2 [1].

### 2.6.3. Gunning Fog Index Formula

This is a readability formula used to measure English text readability and generate scores, which indicate the level of education year required to understand the text as shown in Table 3 [6,32].

**Table 3.** Gunning's fog-index level [31].

| Fog Index | Reading Level by Grade |
| --- | --- |
| 20+ | Post-graduate plus |
| 17–20 | Post-graduate |
| 16 | College senior |
| 15, 14, 13 | College junior, sophomore, freshman |
| 11–12 | High school senior, junior |
| 10 | High school sophomore |
| 9 | High school freshman |
| 8 | 8th grade |
| 7 | 7th grade |
| 6 | 6th grade |

The following is the algorithm to determine the Gunning fog index.

✓ Calculate the average number of words used per sentence.
✓ Calculate the percentage of difficult words in the sample (words with three or more syllables).
✓ Add the totals together, and multiply the sum by 0.4.
✓ Algorithm:(average_words_sentence + number_words_three_syllables_plus) × 0.4

*2.7. Source Code Comments Assessment*

In the coding area, many tools used previous formulas or create their own formula. An approach for quality analysis and assessment of code comments by [8]. The provided approach defined a model based on categories of the comments. Researchers applied a machine learning technique on the developed application which is programmed in Java and C/C++. Authors used a metric to evaluate the coherence between codes and comments of methods, for example, the name of the routine and its related comment. In addition, they used another metric that investigates the length of experimented comments. As for coherence, authors compare the words in comments with ones that founded in the method name. Moreover, the aim of using the length of the comment is coming from the assumption that the short inline comment may contain less information compared with long ones. To apply this study, they use surveys that distributed over 16 experienced software developers. This work is related to our work by assessing the source code comment, but authors do not care about the readability level of comments as written text; they only care if there is a relation between the code and the comment itself. However, in our research, we focus on readability level of comments and its words completion to achieve the purpose of existence of these comments [26,32].

The authors of [33], in their research collected methods for java programs from six popular open-source applications. They applied analyses on comments from collected datasets; to do this they conducted two preliminary studies on words appearing in comments and on amounts of comments. The results demonstrated that most of the comments were categorized as short sentences that contain at most 10 words. Besides, the methods that inner code has more lines of comments could need more changes in the feature. Therefore, it would require more time to fix, especially after the product is released as a production version. This result may conflict with the work that we can use good comments besides good source code. In addition, if these comments are not as the user expected, we can improve the readability without affecting the code quality itself.

According to [19], the researchers developed the Javadoc Miner tool to assess the quality of one type of comment, which is in-line documentation by using a set of heuristics. To assess the quality of language and consistency between source code and its related comments. Authors measure

the readability of comments by assessing the quality of language that comments were written with heuristics by counting the number of tokens, nouns, and verbs, calculating the average number of words, or counting the number of abbreviations. In addition, they used the fog index or the Flesch reading ease level to assess the readability level of comment text. The main aim of authors in this research is to detect inconsistencies between code and comments, by checking all properties of methods and even these properties documented in comments and explained as others can understand it. Authors found that the comments are not up to date which caused misunderstanding in the working of these methods. In addition, authors noticed that the codes which are well commented have less faults or problems reported than ones that have a bad comment that has more fault and bugs.

Researchers in [34] were created two data sets from tow corpora which were Penn Discourse Treebank and the Simple English Wikipedia corpora to be used as a sample in their research and apply the researched feature that used to assess the complexity of the text. These features were divided into five groups as surface, lexical, syntactic, cohesion and coherence features. They found that coherence features are needed to be in combination with others and if these features dropped from combination there is a significant decrease in accuracy, this led to result as there is a strong correlation between text coherence and text complexity [35].

Researchers of [33] amid to prove the relationship between the fault-proneness and the commenting manner in methods declared in Java classes. They focus on two types of comments which were: documentation comments and inner comments. To achieve their aim they used two methods (Analysis-1 and Analysis-2). The results of this research were that a function with inner comments is faultier than a non-commented method; in addition, using comments may indicate that programmers write poor code or faulty code.

### 2.7.1. Proposed Methodology of Code Comments Evaluation

The main goal of our research is to enhance code comments readability, therefore to make understanding the code and related works processes (maintainability, reusability, and reviewing) easier and this will achieve the purpose of the existence of the comments. We go into building CRS "comments readability system" as a tool to be used by programmers to verify the readability of their comments in the development phase and suggest improvements to enhance the comments readability level to be more understandable and valuable to anyone who will be in process of using this code. This tool will use three formulas to check the current comment complexity level, by this immediate evaluation, new terms or words will be suggested to the programmer. The Flesch Reading and Gunning fog formulas with modification of their measuring parameters beside database of the suggested complex word will be used to assess the writing comment. Then each word that does not satisfy the preconfigured criteria will be marked and scored. This will provide an opportunity for the programmer to go through each marked word and provide readable alternatives from the database or corpus, which will be directly connected to the tool. The prosed system has been evaluated using GitHub data, which was collected manually as we search for open source project and get the comment in file to put them in our system to check if we can change the readability of these comments. In addition, we used a survey, which was distributed through email, and form was created on Google form. Therefore, we sent the link to many companies and also to many student groups, especially in computer science, within Birzeit University club groups.

### 2.7.2. The Proposed Approach

The proposed system used consists of two modules, the measurement readability module, and the replacement words module. As the following Figures 2 and 3 show, the general processes that are executed in each phase of comment text readability measurement and alternative terms suggestion.

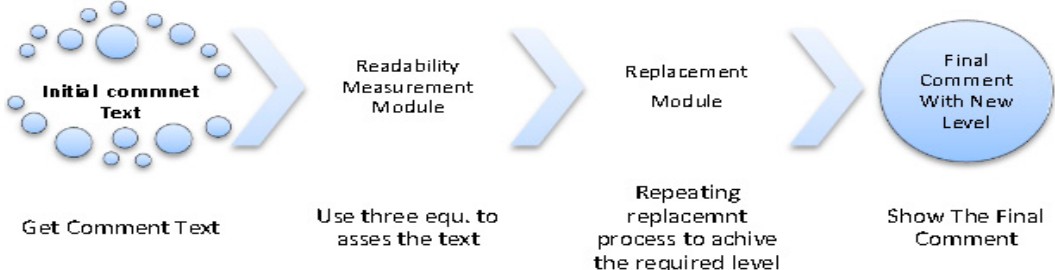

**Figure 2.** System general steps.

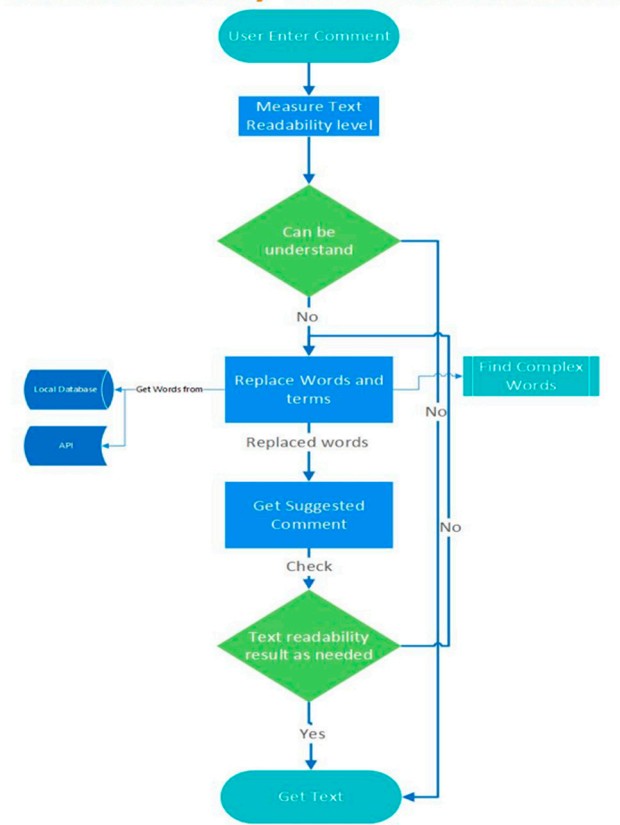

**Figure 3.** Overall system process.

In each part of the system, there is a sub-process executed to get to the final stage the required text with simple words and terms the following diagram. The overall system process shows how text passes through system modules and what is happing in each stage.

The main two modules that the system consists of are measurement readability and the replacement module. The following two sections explain what is happening at each module stage.

The measurement readability module is used to measure the comment text readability level by using three equations from the three formulas used in this research. To assess the readability level for a text of target comment and extend the complex words from the text and set those as recommended words that should be replaced. This replacement module is used to replace the suggested word from a local database or an online database consumed from Application Programming Interface (API). The replacement term retrieved locally was replaced automatically but when online it gives the user a list of suggestions selected manually, and for which terms are more readable for him/her. On the other hand, the listed term is scored by API teams to show the most suitable term for the requested word as semantic or as generally used in daily life. By combining these two ways we will get more options

for the current text to determine the best alternative word, which gives other people the chance to understand what the writer means from these comments.

　　　Figure 3 shows the text readability enhancement process which will be applied to generate the proposed system. The readability measurement module is considered the heart of the proposed tool. It depends on three formulas to measure the level of text readability (fog index, Flesch reading-ease, Flesch–Kincaid). Figure 4 shows the internal process done from entering text, and checking readability, from three formulas as functions. Finally, we get the level of readability as a number with a listing of complex words that could be replaced with simple ones.

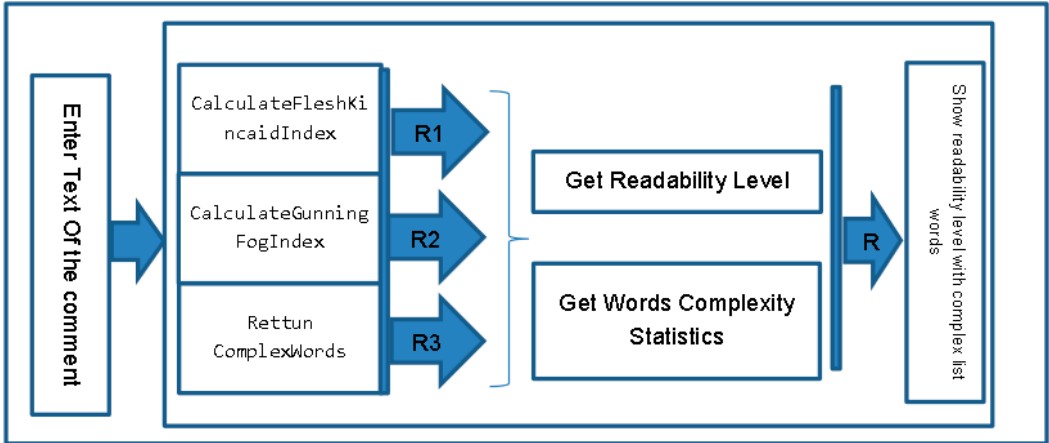

**Figure 4.** Module functions and processes.

　　　Each calculation formula is created as an individual function called from the main screen, in addition to supported functions used to extract entrances and extract words from each sentence.

　　　The following code for the fog index formula calculated:

Dim indexval As Double = 0.4 × ((CDbl(wordcount)/sentencecount) + 100 * (CDbl(complexwords)/wordcount))

　　　As we see, the formula depends on word count, the count of sentences, complex word (this part depends on the count of syllables where we use three and more), and the word count in the submitted text. The score as we mentioned before, while it increased the readability also increased.

　　　The Flesck Incaid index is calculated using the following formula:

Dim indexval As Double = 0.39 × ((CDbl(wordcount)/sentencecount) + 11.8 × (CDbl(syllablescount)/wordcount) − 15.59)

　　　This formula depends on different parameters in which fog depends on which is syllables count.

　　　Furthermore, there is a SyllableCount function, which is used to get the complex words, in addition to the syllable count, and return the complex word into an array of words to be changed after the whole function and process is done for this phase of the module.

　　　By way of the measurement process, the result of text ratability can be evaluated and determined. The complex words that should be replaced are identified to make the text more readable therefore more understandable.

### 2.7.3. An Example of System Use

　　　The following Figure 5 shows the tool main screen, the screen has many parameters with two windows: the first one is "text body". The text which is written by programmer "comment" context, each complex word will be highlighted to indicate that the written word is classified as complex and draws programmer's attention to change it. The second window is "suggested text" that suggests replacing some of the words found to be complex or if there is a simple alternative for it from the local

database. Figure 6 has many labels that display the result of measuring readability for the formula used to measure text readability. In addition, there is a list of word complexes in front of it, and there is a list of words that have been changed in the suggested text.

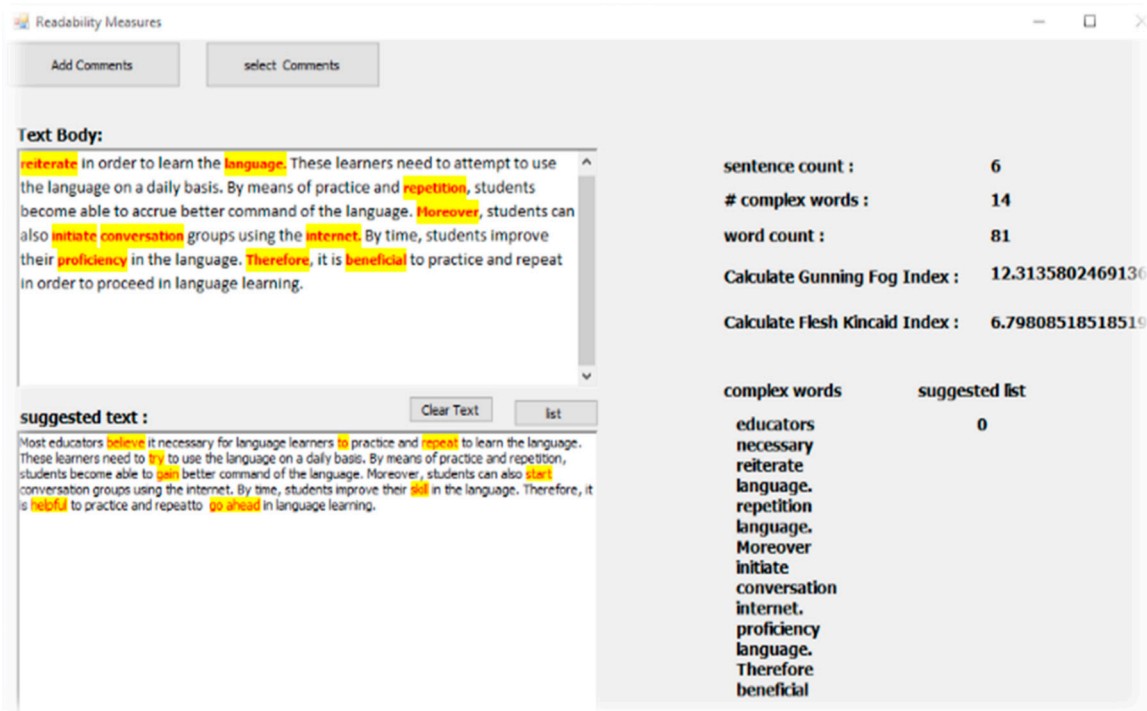

**Figure 5.** Main screen of the system.

sentence count :  6

\# complex words :  14

word count :  81

Calculate Gunning Fog Index :  12.3135802469136

Calculate Flesh Kincaid Index :  6.79808518518519

**Figure 6.** System calculations section view.

Figure 7 shows how the readability level will be displayed to the end-user. It listed the complex words that should be replaced as it's measured as complex words and the number of syllables more than the threshold values.

**complex words**

educators
necessary
reiterate
language.
repetition
language.
Moreover
initiate
conversation
internet.
proficiency
language.
Therefore
beneficial

**Figure 7.** System complex words list view.

In the replacement module, we care about the words used to be a substitution for original ones. Further, we make sure the term is not complex and usually used. For this reason, we used both local and online databases for replacement. In this section, we emphasize the main functions used to get the best result and be sure the enhancing of the readability level is as we imagine.

The local database consists of many complex words with simple alternative words collected from internet web sites. Table 4 shows a sample of it.

**Table 4.** Sample complex words with a simple alternative.

| Prime | Alternative |
| --- | --- |
| accrue | gain |
| adjacent to | next to |
| advantageous | helpful |
| allocate | divide |
| apparent | dear |
| ascertain | learn |
| attempted | tried |
| beneficial | helpful |
| capability | ability |
| category | group |
| close proximity | near |
| commence | start |
| comply with | follow |
| component | part |
| concur | agree |
| consolidate | combine |
| constitutes | forms |
| convene | meet |

The functions of the changing words loop through the text of the comment, and replacing all words found with an alternative without user intervention. These terms come from the local library, which will be used as the local corpus, as shown in Figure 8.

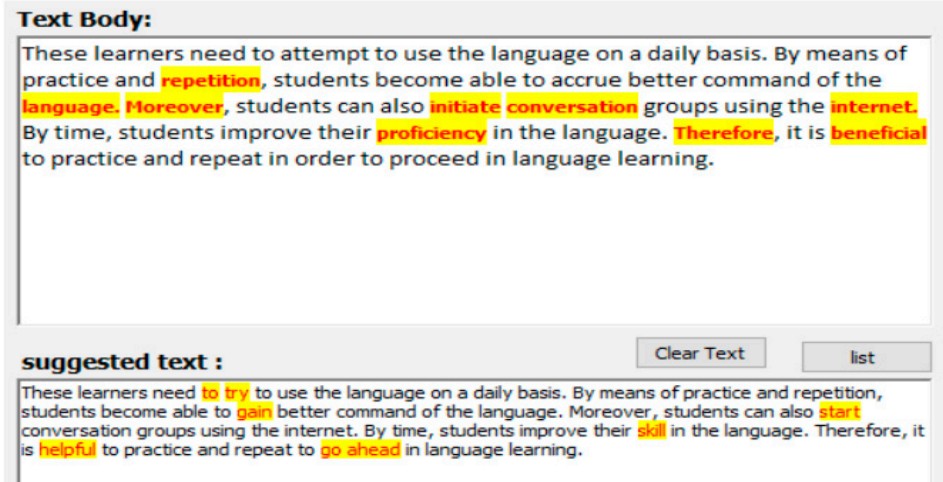

**Figure 8.** Example of using a local database for the replacement of the terms.

Each word replaced from the original text is highlighted as yellow with changing the color to red in both text boxes (text body, suggested text).

API is another source of word replacement; this source returns a list of suggested words from API service consumed by sending the word that we need to replace or check whether this word has an alternative simple term.

The list of words returned as JSON are listed with two fields; the first is the alternative word; the second is the score. For this, I quote "for queries that have a semantic constraint, results are ordered by an estimate of the strength of the relationship, most to least. Otherwise, queries are ranked by an estimate of the popularity of the word in written text, most to least. At this time, the "score" field has no interpretable meaning, other than as a way to rank the results" [36].

We used the function "getlist" to consume the API and return a list in the dataset as JSON.

When selecting any word from the text body, the following screen will appear, i.e., the word we want to change is "initiate" the list as shown in Figure 9 below:

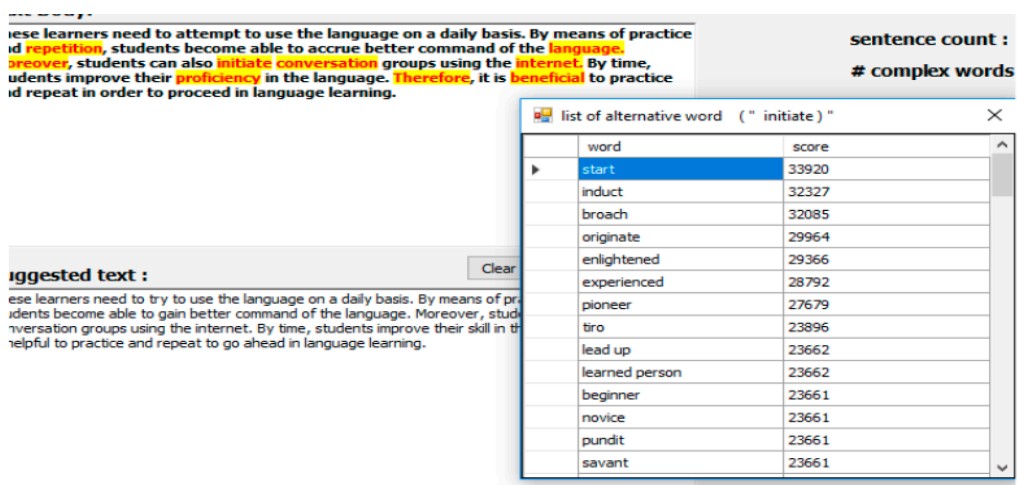

**Figure 9.** Alternative words for "initiate".

After looping in the text body and getting new text in the suggested text, we can get the final text and pass it into the text body text to check the readability level.

## 3. Results

### 3.1. Results of Software Testing

Testing is an important phase in developing and our research to help testers and Quality Assurance staff (QAs) to complete their tasks with a short time. In addition, with less effort, the benefit from the existence of the comments and the documentation of the code is realized as their English readability level.

For this phase, we use the text as the testing paragraph written by Dr. Majdi Abu Zahra from the English and translation department in BirZiet University (BZU). This paragraph was tested by our proposed tool as shown in Figure 10 and was retested by online free calculated text readability using fox index as Figure 11 shows the result which shows that the indexing value was close to each other.

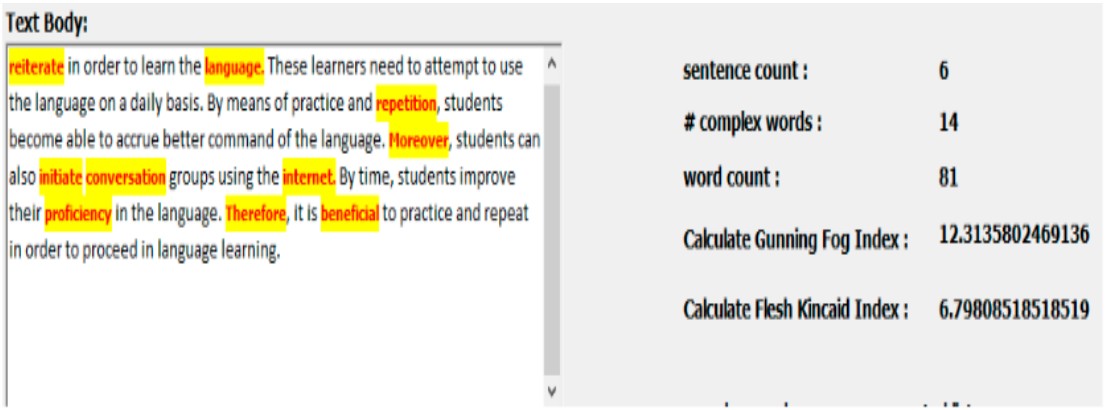

**Figure 10.** Testing tool result.

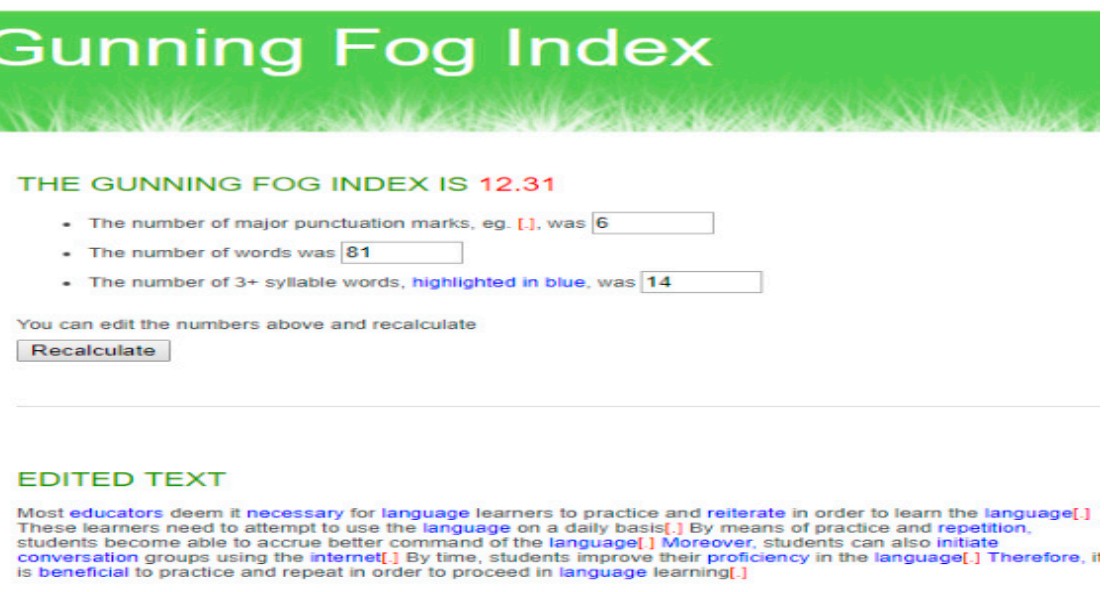

**Figure 11.** Testing from another resource: http://gunning-fog-index.com/fog.cgi.

### 3.2. Results of Conducted Survey

#### 3.2.1. Results of Survey with Programmers

Within this scope, the result from the survey was that there are four main languages that developer set as main programming languages: Java and its percentage was 17.1%, PHP with a percentage of 19.5%, Python with a percentage of 12.2%, and .Net with a percentage of 24.4%. The rest value for 10% language as a percentage of 2.4% to 5% for each as shown in Figure 12.

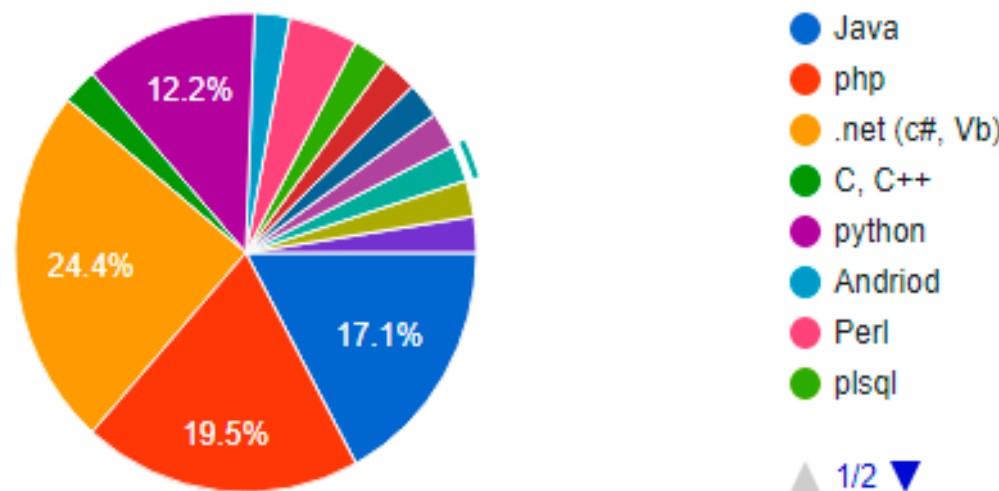

**Figure 12.** Result for the primary programming language of participants.

The following charts (Figure 13), show that most of the programmers have a very high English level. This is displayed in three parts (reading, writing, and speaking).

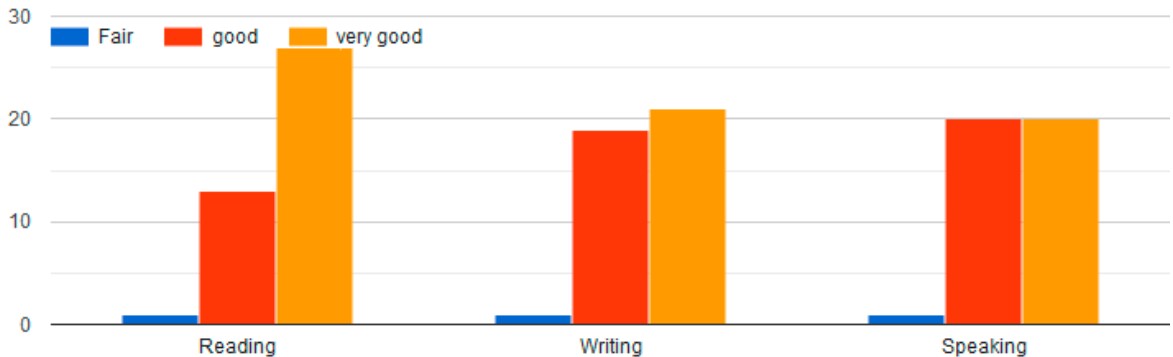

**Figure 13.** Programmer English level.

The result relating to working in a group was 90.24%, as shown in Figure 14, which means that most of them preferred to work in a group rather than work individually.

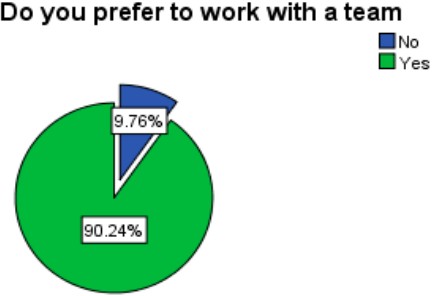

**Figure 14.** Working in a team result.

In addition, the results about the comments are shown (Did you use comments in coding? Do you think the comments in the code are necessary? And do code comments help you in understanding the code?). After reviewing the overall result, we found that most programmers encourage others to use comments and they support the existence of comments in a code file, as shown in Figure 15.

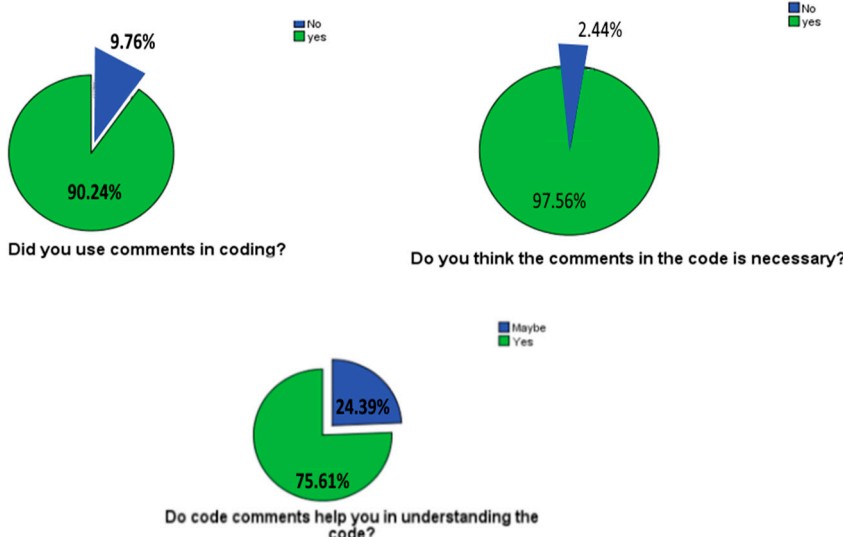

**Figure 15.** Result about source code comments.

However, the important part of the survey is about the three questions asked about the original comments which were collected from GitHub projects and the new one, which was the result of our proposed tool. The chart below shows the first question about which of the two comments: "Has more complex words". The results are shown in Figure 16.

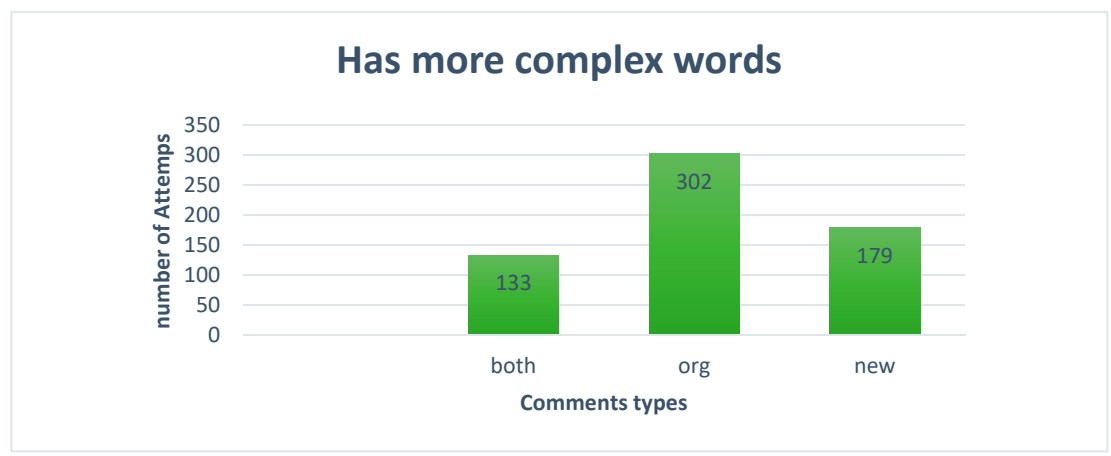

**Figure 16.** The result of the question: which of the two comments has more complex words? Where "org" is the original comment from the sample.

From Figure 16, the replacement function with alternative words makes text words simpler, and the result was that the original comments have complex words marked as 49.11%. However, the new one had a percentage of 29.11%. This means that there a differences in complexity between the two texts as the results show. Further, this may affect the time required for reading the text of the comments. In addition, the result showed a positive relationship so the factor of a number of complex words takes more time to read and the reader gets the main idea from the text. Figure 17 below shows the result of the time taken to read the text of each comment. Programmers see that the text of original comments, which were mentioned before as having more complex words, took more time to read.

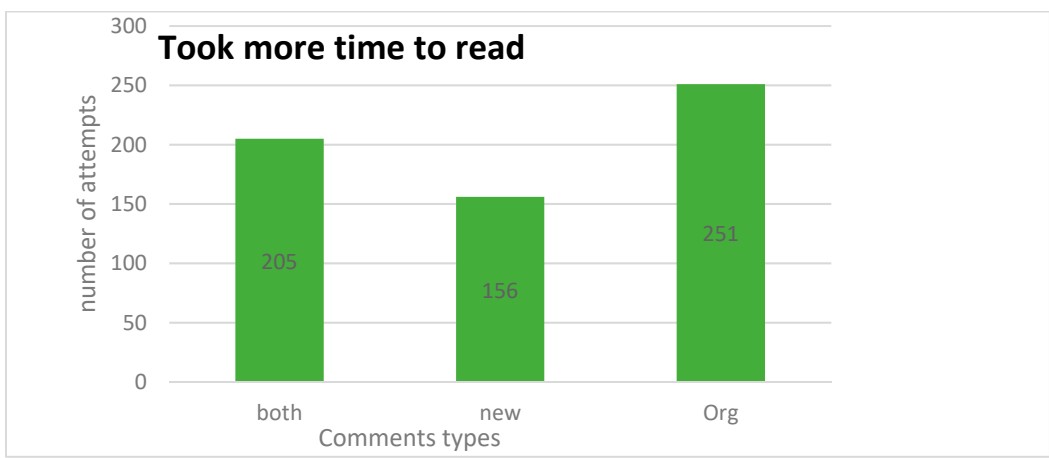

**Figure 17.** Result of question: which took more time to read? Where "org" is an original comment from the sample.

Finally, about the last question which was: "Which one was more understandable?" The results for the two texts were within the same level, and this is because the advanced level of English and the complexity of words did not affect the understandability of reading because these programmers are proficient in English, as we discussed earlier, and this assumption has resulted in [23,37] saying that there a strong correlation between reading comprehension and vocabulary knowledge. Figure 18 below shows this.

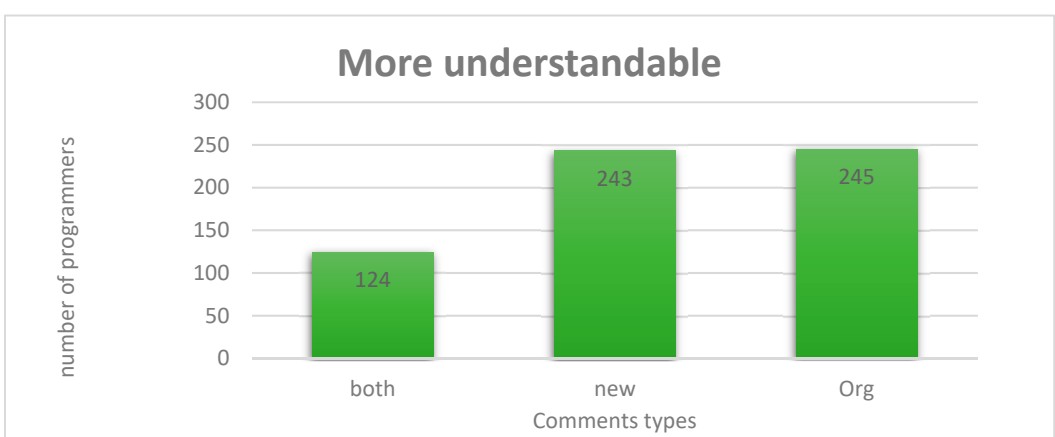

**Figure 18.** The result for question: which was more understandable? Where "org" is the original comment from the sample.

3.2.2. Results of Survey with Students

Birzeit University students from the Faculty of Engineering and Information Technology, as mentioned in the Section 3.2 were surveyed via random selection. The follow results were obtained from Computer science students with year levels of 3, 4, and 5. The results, expressed as year level, are shown below in Figure 19:

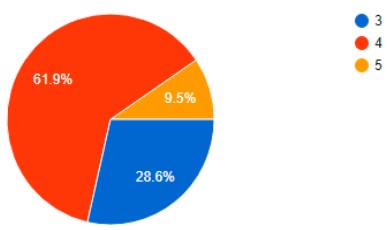

**Figure 19.** Student year level.

However, we now turn back to the important part of the survey about the three questions asked about the original comments, (which were collected from GitHub projects and the new ones, which were the result from our proposed tool). From the first question about which of two comments "has more complex words", the result is shown in Figure 20.

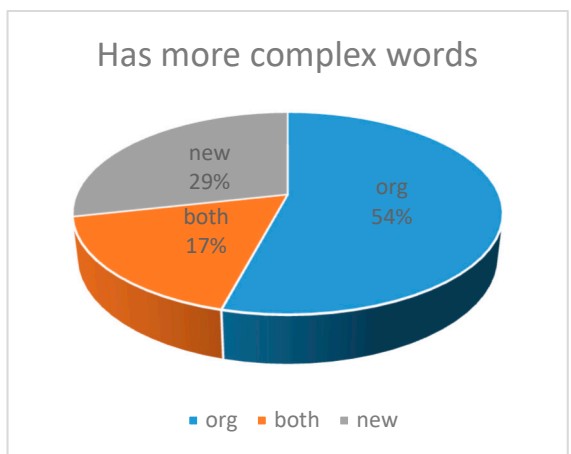

**Figure 20.** The result of the question: which of the two comments has more complex words? Where "org" is the original comment from the sample.

From Figure 20 above, the replacement function with alternative words makes text words simpler and the result was as the original comments have complex words marked as 54%. Whereas the new was with a percentage of 29%; this means that there is a difference in complexity between the two texts as result have shown. Further, this may affect the time of reading the text of the comments. In addition, the result was a positive relationship that the factor of a number of complex words took more time to read and get the main idea from the text. Figure 21 below shows the result of the time taken to read the text of each comment. Students believe that the text of original comments mentioned before has more complex words which took more time to read; the result of this finding was the new comments took less time to read with a percentage of 23%, whereas the original was with a percentage of 40%.

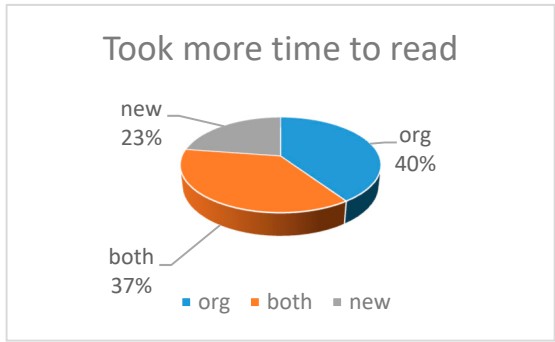

**Figure 21.** Result of question: which took more time to read? Where "org" is an original comment from the sample.

Finally, in regards to the last question which was: "which one was more understandable?", Figure 22 shows that the results almost showed that the new comments were more understandable with a percentage of 56%, but the original comments were 30%. This gives an indicator that the level of English and experience may affect readability.

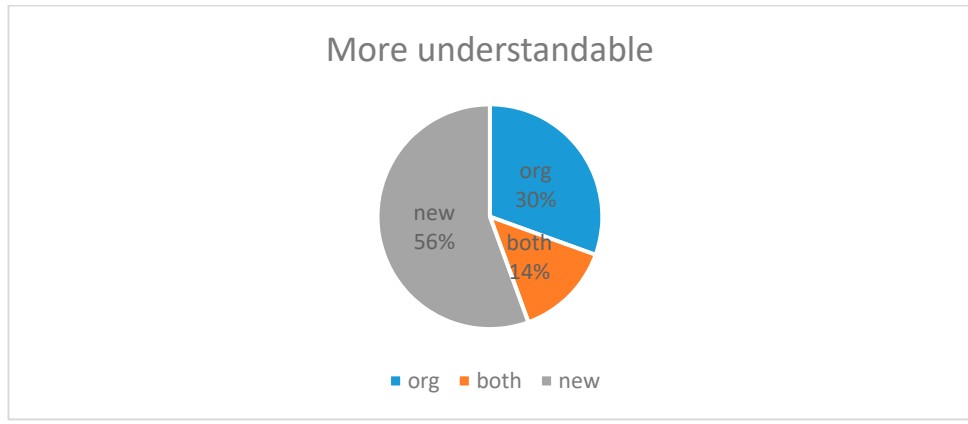

**Figure 22.** Result for the question: which was more understandable? Where "org" is an original comment from the sample.

## 4. Discussion

For programmers to do data analysis we apply a *t*-test on data to approved research hypotheses measuring readability. The result is shown in Table 5. With a 95% confidence interval of the difference, there is no significant difference for an understanding factor of new text compared with original text because the value was (0.969) and this value is more than 0.05. Therefore, this result may not be effected by text complexity because of the high English level of programmers in our random sample. However, the result of the variable of text complexity (text with more complex words) was (0.00) which is less than (0.05); this means that there is a significant difference between the original text and the new text. Furthermore, this evidence supported the research hypothesis that the CRS (with the replacing module) reduces text complexity level and thus the *t* value was −4.458; which means the new text contains fewer complex words than the original one. Finally, the last measured variable was the time taken to read the text. The result value was (0.01) and this is less than 0.05 which means that there is a significant difference between compared comments; and also the *t* value is −3.788, which means that the new text took less time to read than the original one. Thus, this supported our assumption about text readability to reduce the time to understand the comments on the source code. Finally, we can say there is a significant difference in comment readability level using CRS tool in the development phase and comment readability level without using CRS. This significance was that the new one that contains a smaller number of complex words and took less time to read.

**Table 5.** Programmers data—new comment readability *t*-test values.

| One Sample Statistics | | | | |
|---|---|---|---|---|
| **Evaluation Criteria** | **N** | **Mean** | **Std. Deviation** | **Std. Error Mean** |
| more_understandable | 41 | 0.0016 | 0.26977 | 0.04213 |
| more_complex | 41 | −0.2016 | 0.28963 | 0.04523 |
| more_time_to_read | 41 | −0.1512 | 0.25561 | 0.03992 |

| One-Sample Test | | | | | | |
|---|---|---|---|---|---|---|
| | | | Test Value = 0 | | | |
| **Evaluation Criteria** | **t** | **df** | **Sig. (2-tailed)** | **Mean Difference** | **95% Confidence Interval of the Difference** | |
| | | | | | **Lower** | **Upper** |
| more_understandable | 0.039 | 40 | 0.969 | 0.00163 | −0.0835 | 0.0868 |
| more_complex | −4.458 | 40 | 0.000 | −0.20163 | −0.2930 | −0.1102 |
| more_time_to_read | −3.788 | 40 | 0.001 | −0.15122 | −0.2319 | −0.0705 |

In addition, we applied the test correlation type between three questions to check if our hypothesis is true. As the table below shows, there is a relation between the three variables we search for (number of complex words, text understanding, and the time taken to understand the whole text and get the idea from it). The results support the research and assumes that the text with more complex words takes more time to read, and vice versa. In addition, Table 6 shows each variable compared to the other two variables. If we study the relationship between understanding text and containing a number of complex words, the relation was revised (more understandable for the less complex words, and vice versa). On the other hand, more time taken to read means that the text contains more complex words and more time means less understanding, which is what the study is trying to prove.

**Table 6.** Results from the correlation between the three main questions.

| | | Correlation | | |
|---|---|---|---|---|
| | | More_Complex | More_Understandable | More_Time_to_Read |
| more_complex | Pearson correlation | 1 | −0.674 ** | 0.532 ** |
| | Sig. (1-tailed) | | 0.000 | 0.000 |
| | N | 41 | 41 | 41 |
| more_understandable | Pearson Correlation | −0.674 ** | 1 | −0.497 ** |
| | Sig. (1-tailed) | 0.000 | | 0.000 |
| | N | 41 | 41 | 41 |
| more_time_to_read | Pearson Correlation | 0.532 ** | −0.497 ** | 1 |
| | Sig. (1-tailed) | 0.000 | 0.000 | |
| | N | 41 | 41 | 41 |

** Correlation is significant at the 0.01 level (1-tailed).

For students' data analysis, we apply a *t*-test on the data to prove the research hypothesis regarding measuring readability. The result, as shown in Table 7, with a 95% confidence interval of the difference, shows that there is a significant difference for the understanding variable of new text compared with original text because the value was (0.000) and this value less than (0.05). In addition, the t value was (5.066) which means that the new comment is more understandable than the original one. Another variable was text complexity (text with more complex words). The result was (0.00) which is less than (0.05) and this means that there is a significant difference between the original text and the new text. Furthermore, this evidence supported the research hypothesis that the CRS (with the replacing module) reduces text complexity level as the t value was (−4.695), which means the new text contains fewer complex words than the original one. Finally, the last measured variable was the time that it took to read the text; the result value was (0.01) and this is less than 0.05, which means there is a significant difference between compared comments. Furthermore, the *t* value is −3.837 which means that the new text took less time to read than the original one. Thus, this supported our assumption regarding text readability as a way to reduce the time to understand the comments on the source code. Finally, we can say there is a significant difference in comment readability level using the CRS tool in the development phase and comment readability level without using CRS. This is a significant finding showing that the new one is more readable than the original text comments.

**Table 7.** Student data—new comment readability *t*-test values.

| One Sample Statistics | | | | |
|---|---|---|---|---|
| **Evaluation Criteria** | **N** | **Mean** | **Std. Deviation** | **Std. Error Mean** |
| new_has_more_complex words | 35 | −0.2438 | 0.30719 | 0.05192 |
| new_has_more_understandable | 35 | 0.2343 | 0.27362 | 0.04625 |
| new_has_more_time_to_read | 35 | −0.1695 | 0.26139 | 0.04418 |

| One-Sample Test | | | | | | |
|---|---|---|---|---|---|---|
| | Test Value = 0 | | | | | |
| **Evaluation Criteria** | **t** | **df** | **Sig. (2-Tailed)** | **Mean Difference** | **95% Confidence Interval of the Difference** | |
| | | | | | **Lower** | **Upper** |
| new_has_more_complex words | −4.695 | 34 | 0.000 | −0.24381 | −0.3493 | −0.1383 |
| new_has_more_understandable | 5.066 | 34 | 0.000 | 0.23429 | 0.1403 | 0.3283 |
| new_has_more_time_to_read | −3.837 | 34 | 0.001 | −0.16952 | −0.2593 | −0.0797 |

In addition, we apply the test correlation type between three questions to check if our hypothesis is true. As the table below shows there is a significant relationship between the three variables we search for (number of complex words, text understanding, and the time that takes to understand the whole of text and get the idea from it) because correlation is significant at the 0.01 level. The results that support the research assume that the text with more complex words takes more time to read and also the about understanding it was an inverse relationship. In addition, Table 8 shows each variable with the other two variables. If we study the relation between understanding text and containing a number of complex words, the relation was a revised relation (more understandable with the less complex words, and vice versa). On the other hand, more time to read means the text contains more complex words and thus also leads to less understanding, and this is what this research attempts to prove.

**Table 8.** Result of the correlation between three main questions.

| Descriptive Statistics | | | |
|---|---|---|---|
| **Evaluation Criteria** | **Mean** | **Std. Deviation** | **N** |
| new_has_more_complex words | −0.2438 | 0.30719 | 35 |
| new_has_more_understandable | 0.2343 | 0.27362 | 35 |
| new_has_more_time_to_read | −0.1695 | 0.26139 | 35 |

| Correlations | | new_has_more_ complex words | new_has_more_ understandable | new_has_more_ time_to_read |
|---|---|---|---|---|
| new_has_more_complex words | Pearson Correlation | 1 | −0.699 ** | 0.639 ** |
| | Sig. (2-tailed) | | 0.000 | 0.000 |
| | N | 35 | 35 | 35 |
| new_has_more_understandable | Pearson Correlation | −0.699 ** | 1 | −0.644 ** |
| | Sig. (2-tailed) | 35 | | 0.000 |
| | N | 1 | 35 | 35 |
| new_has_more_time_to_read | Pearson Correlation | 0.639 ** | −0.644 ** | 1 |
| | Sig. (2-tailed) | 0.000 | 0.000 | |
| | N | 35 | 35 | 35 |

** Correlation is significant at the 0.01 level (2-tailed).

## 5. Conclusions and Future Works

### 5.1. Conclusions

The quality code depends on many factors that influence its readability; one of them is the comment (the part which describes the code and gives more information about the section which is written about). Therefore, this comment is written to be used by other developers and developers who write the code, for this issue these comments should be readable and easy to be understood by others. Therefore, we will measure the readability of the comments and make some changes. By this change, we aim to make its readability level suitable for others. These changes will contribute to achieving the target readability level, and they will also make understanding code easier. Code understanding means that code readability is easy, and therefore, maintainability and reusability become easier also.

We implement a tool to assess source code comments readability, this tool used three famous formulas (fog index, Flesch reading ease score, and Flesch–Kincaid grade level) to measure text readability, also used two resources for words and terms alternatives. We used an updated API to be sure that the alternative word choices were updated and closer to the required terms. We collected comments from Github projects then reviewed them by an instructor at the Department of Languages and Translation at BZU—Dr. TAWFIQ Ammar. All of these comments that have complex and unusual words were replaced by terms with less complexity and are familiar to most people. However, this change affected the readability level for each of the comments. This was evaluated by creating a survey completed by 41 senior programmers, in addition to 35 students. The survey consisted of two sections: the first one was about general information about the experimenters. In addition, the second one consisted of 15 questions, each question contained two comments—one from Github projects and another one was from the enhanced comment generated by our proposed tool. The experimenter then answered which of two comments had more complex words, had more understandable words, and which one took more time to read. Survey participants (programmers) agreed that enhanced comments contained a less number of complexwords as percentages—449% for original comments, 29% new comments which had more complex words, and 22% where there was no difference. Further, these comments were more understandable than the original ones—40% for original comments, 40% for new comments which had more complex words, and 20% there is no difference. Finally, enhanced comments took less time to read as percentages—41% for original comments, 25% for new comments which had more complex words, and 34% where there was no difference. With previous information from t-test analyses, the result showed that the new comments were better in two variables—the new comments contained more complex words, and the new comments took more time to read. However, the result understated that there was no difference between the two compared comments. We guess that this was with the high level of English of expert programmers, where the complex words did not affect the understanding of the text.

For survey participants who were students—they agreed that enhanced comments contained a smaller number of complex words as percentages—54% for original comments, 29% for new comments that had more complex words, and 17% where there was no difference. In addition, these comments were more understandable than the original ones—30% for original comments, 56% for new comments which had more complex words, and 14% for where there was no difference. Finally, enhanced comments took less time to read—40% for original comments, 23% for new comments which had more complex words, and 37% for where there was no difference. With previous information from *t*-test analyses, the results show that the new comments were better in three concerning variables (new and requires more understanding, is new and contains more complex words, and where the new comment took more time to read), which were different from programmers where the understanding of the text was not affected. We assume, as mentioned before, that this was due to the high level of English language used.

From previous discussions, we can say that the tool which we built may help people for whom English is not their first language, rather than those who have a very high level of English, or those who are native speakers of English.

*5.2. Future Works*

In the future, many upgrades can be done to this system:

i   Add this tool as an add on to Integrated Development Environment (IDE); this will make reading comments easier and help programmers to change the complex word while writing their own comments.

ii  Connected tool to corpus with other languages. Because we focus on English and make alternative words from the list retrieved from this corpus.

iii Add spellchecking that may help programmers to reduce the error that comes up from typo mistakes.

iv  Add more terms to local languages. That should be used in the future as a local corpus.

v   Add Natural Language Processing (NLP) smart metrics to change terms with suitable alternative terms.

vi  Add new features to consume the comments automatically from projects and combined as enhanced documentation.

**Author Contributions:** D.E. proposed the idea and provided some of the literature review and related works. A.O. implemented the experiment, generated the proposed system, conducted the survey and analyzed the data. A.E. contributed to writing, verifying and validating the results as well as proof reading the article. All authors have read and agreed to the published version of the manuscript.

**Funding:** This research received no external funding.

**Conflicts of Interest:** The authors declare no conflict of interest.

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
