# Peer review of "Enhancing Software Comments Readability Using Flesch Reading Ease Score"

_information, doi:10.3390/info11090430_

Round 1
Reviewer 1 Report
The authors of this research claims that on of the factors affecting the quality of source code comments is readability. They propose a comments readability system be used by programmers to verify the readability of their comments and suggest improvements to enhance the comments readability level to be more understandable and valuable. Authors use three formulas to measure the level of text readability: fog index, Flesch reading-ease and Flesch-396 Kincaid. In general, the manuscript requires a re-structure to follow IMRAD. Suggested revisions:
- Literature review section is too long and contains also theoretical background.
- Methodology section contains also results.
- Result section is missing.
- Check arrows in Figure 5.
- Figure 7 does not have any added value.
- Figure 12 does not have any added value.
- Proof read English in all the manuscript.
Author Response
- Methodology section contains also results.
Results has been removed to anew created section
- Result section is missing.
result section has been created
- Check arrows in Figure 5.
It is checked and now it is OK
- Literature review section is too long and contains also theoretical background.
Literature review was revised and shortened by amend not relevant literature and concise elaboration.
- Figure 7 does not have any added value.
Figure 7 removed.
- Figure 12 does not have any added value.
Figure 10 removed.
- Proof read English in all the manuscript.
Proof read has been performed.
please the full paper with track changes.

Reviewer 2 Report
Interesting research that shows practical usefulness and scientific foundation of results and conclusions. Authors show deep understanding of the problem. They presented detailed analysis of related work, different methods and tools application and sensitiveness for the need of real-world practice, specially for the professionals and students population needs. It was really a great effort and detailed approach to the scientific research methodology application presented by authors in this work. The work presented in this paper is easy-readable text with clear presentation and use of presentation elements, such as activity diagram and other visualizations, which enhance understandability of the approach.
What I have noticed that needs to be changed:
- Figure 1 - shows "popular" and "funny" example of comments...Not suitable for a scientific journal.
- Figure 2 - presents "iron triangle" of project management (PM), not business risk. Seems that here project management "iron triangle" was applied to risk management, which is one of 10 knowledge areas in PM. It is true that these factors in "iron triangle" affect project success and if the business is organized with project management basis, then surely these factors are influencing in business success generally. What I suggest here is to have Figure 2. related to PM as well.
- First section in Literature review has title "Software quality code". Authors should be careful with the English formulation. Better: "Software code quality". Second remark here is that this part is related to software quality in general, not the software code quality precisely, so it would be better to adjust the title to correspond the content.
- Readability formulas presented in Literature review section, that could be used for comments, are not clear about how could be applied. They have been described with partial presentation of formula elements and formulas that use these elements(factors) combined with specific constant values. It is not clear why these constants are used (e.g. 1.015).
- Methodology section introduce more precise formula to be included in calculations, again with constants such as 0.4 (fog index)...Why this 0.4? Flesck - 0,39, 11,8?
- Authors should separate research methodology from the methodology of estimating quality of comments that is included in software, as well to make distinction with separate section precisely related to the developed tool that apply the selected methodology. It is not good to have all these 3 together in one section (research methodology, estimation methodology and software tool). Having all these together makes the reader of this paper confused a bit. It is advicable to have the following structure of the whole paper: Introduction, Theoretical Background (not supported - to explain basic terms and definitions), Literature Review, The comments quality measurement methodology, The evaluation tool, The research methodology (not supported - to have explained how the research is conducted with population, what is reseach population - programmers, students, where were they contacted, how, questionnaire structure, was the tool used by research population and how, or only survey has been conducted without the use of the previously introduced tool, if the tool was used, what is the research sample of software - shortly mentioned Github projects, but which ones...were they used with code inspection&survey comparing with the same projects analyzed with the proposed tool), Research results, Conclusions and Future work.
General conclusion: Even the paper has presented interesting topic and detailed approach, there are still certain need for having better organization of the whole text and more details especially in the research methodology section.
Author Response
- Figure 1 - shows "popular" and "funny" example of comments...Not suitable for a scientific journal.
It is removed
2. Figure 2 - presents "iron triangle" of project management (PM), not business risk. Seems that here project management "iron triangle" was applied to risk management, which is one of 10 knowledge areas in PM. It is true that these factors in "iron triangle" affect project success and if the business is organised with project management basis, then surely these factors are influencing in business success generally. What I suggest here is to have Figure 2. related to PM as well.
I prefer to keep it.
4. Readability formulas presented in Literature review section, that could be used for comments, are not clear about how could be applied. They have been described with partial presentation of formula elements and formulas that use these elements(factors) combined with specific constant values. It is not clear why these constants are used (e.g. 1.015).
These constants are related to the formulas, I do not know why they have been used.
5. Methodology section introduce more precise formula to be included in calculations, again with constants such as 0.4 (fog index)...Why this 0.4? Flesck - 0,39, 11,8?
same as above.
please see the full paper with track changes for more details.

Reviewer 3 Report
I believe this work did not receive a strong improvement to justify acceptance.
Important fixes.
Tables must be redone and should never be imported as images. The quality of these elements is too low, moreover, I would suggest keeping a consistent style across the paper that ease the reding.
Graphs and charts must be consistent preserving the same style. The quality is too low in several cases and the reading is not possible when the paper is printed.
Avoid the use of 3D graphs when the third dimension does not add extra information. This makes graphs easy to read.
Figure 25 does help to understand the distribution. Think to use a table instead.
All figures must be revised because the style is not consistent and they make confusion.
A spell check must be performed to clean typos.
Previous review:
The authors propose an automated solution to improve code comment readability by systematically replacing unfamiliar words with more commons. Results achieved by surveying developers and students on improved comments show that revised comments appear more readable than the original version.
I appreciate the idea of improving software understandability by swapping uncommon words with their synonymous, however, the approach adopted in the manuscript seems to lack by different points of view.
1. While "Readability and Understandability: Different Measures of the Textual Complexity of Accounting Narrative" appears into references I cannot find a clear citation in text. I would suggest other important works that treat the topic of readability vs. understandability [1], [2].
My feeling is that the related work section does not clarify the difference between readability and understandability. It is a crucial point to give value to this work, I would suggest to expand it adequately.
2. Code review paragraph into related work appears out of the box. It is not related to code readability, It must be removed.
3. Comment types in related work do not have references to automated solutions that have been created to classify code comments in accordance with their natural language meaning. See for a hint [3] [4].
4. It is not clear on which basis the candidates synonymous are more readable than original. Are those words more common for some native speakers than others? I suspect that from this point of view culture, background, or experience play an important role. How do you control these variables during the survey?
5. Minor suggestion, use only flat graphs, charts, and tables. Do not use decorated or 3D pictures when for example the third dimension does not add information to the context. Simple is better for understandability.
6. It is not clear how questions are posed during the survey. Is there any mechanism to prevent bias?
Are the original and improved text present in different order to prevent that the reader always read a certain category first?
7. With word replacement there is a huge risk that the meaning of a sentence is altered. How do you verify that the meaning of a comment is not alternated? For example, the proposed pairs "allocate" and "divide" seems vague synonymous especially when allocate is used to refer to memory allocation! Similar case for "concur" and "agree", the latter cannot substitute the former when referring shared resources or in general concurrency issues.
8. What is the level of spoken English for surveyed people? Does their background influence the survey?
[1] "Readability and Understandability: Different Measures of the Textual Complexity of Accounting Narrative" Smith M. et. al
[2] "Readability and understandability of andrology questionnaires" Hasan Anıl Atalay et. al
[3] "Classifying code comments in Java software systems" Pascarella L. et. al
[4] Listening to programmers taxonomies and characteristics of comments in operating system code" Padioleau Y et. al
Author Response
- Tables must be redone and should never be imported as images. The quality of these elements is too low, moreover, I would suggest keeping a consistent style across the paper that ease the reading.
Tables have been changes from images to drawn tables
- Graphs and charts must be consistent preserving the same style. The quality is too low in several cases and the reading is not possible when the paper is printed.
style has been unified
- Avoid the use of 3D graphs when the third dimension does not add extra information. This makes graphs easy to read.
it is changes from 3D to 2D.
- Figure 25 does help to understand the distribution. Think to use a table instead.
It is removed
- All figures must be revised because the style is not consistent and they make confusion.
All figures have been revised.
- A spell check must be performed to clean typos.
A spell check has been performed
for more information please see the attached paper with track changes.

Round 2
Reviewer 1 Report
The manuscript has been significantly improved and now warrants publication.
Author Response
The manuscript has been significantly improved and now warrants publication.
thanks for your comments and feedback from the first instance.
Reviewer 2 Report
Authors improved overall structure of the paper to have: Introduction, Literature review (including theoretical background - not separated, but separation is not mandatory), Methodology, Result, Discussion and Future work. Authors also added some subsections, to make the whole text more readable.
What should be done for the final paper version:
- taking care of words capitalization, e.g. Line 76 "A Code review" should be "A code review" ...etc.
- Complete text, i.e. in line 205. there is only "designed by Rudolph to measure"...Rudolph who?
- In Proposed methodology, authors should explain why they selected and combined Flesch Reading Ease with Flesch Kincaid and Gunning Fog Index Formula. How these 3 methods were integrated?
- Authors did not explain the role and impact of weighting factors used in formulas (line 353, 361) that were integrated in software , but they have include formulas with weighting factors as they are introduced by original authors. It would be very useful for readers to know the reasons for all weighting factors (206.835, 84.6...0,39, 11.8, 15.9, 0.4). It is supposed that these data about the roots for these weight factors could be found in the source work from original authors of these formulas. Authors should provide several sentences to clarify why to use these weigh factors in these formulas, according to original authors of formulas...
- Authors have only one Methodology title having contents that should be divided in separate sections. Separating sections will make the text more readable.
- Instead of Methodology, there should be titie: "Proposed methodology of code comments evaluation" with subsections: The proposed approach (including figure 2, the algorithm of the approach with figure 3, the use of formulas lines 350-370), Developed software tool (line 319 - 339, 343 -349) An example of system use (From line 371-423)
- Before result section there should have title "Research methodology" with explanation of methods, tools, sample for both: testing and survey. It is especially important to make clear explanation of data collection related to survey, since there is also GitHub projects data collection mentioned in line 463, so it is not clear enough how the data collection from Github is performed (access to source code, code selection...sample formation) and how the Github data collection is related to survey. Finally, the research methods should briefly mention and explain also the methods for data analysis (demonstrated with results starting from 539...).
- Result section should have title "Results" and subsections: Results of software testing, Results of conducted survey (with another subsections - Results of survey with programmers, Results of survey with students).
- Figure 15 is too small and therefore unreadable - percentage in pies could not be read ...!!!
- Conclusion section includes details about the research methodology (data collection, relation with Gihthub,evaluation from the instructor...) All these details should be provided before results section, in research methodology section. Conclusion should be more concise and to make summary of results and provide motivation with shorter discussion about the advantages and disadvantages of the proposed approach.
Author Response
1. taking care of words capitalization, e.g. Line 76 "A Code review" should be "A code review" ...etc.
reviewed and corrected throughout the text.
2. Complete text, i.e. in line 205. there is only "designed by Rudolph to measure"...Rudolph who?
it is completed
3. In Proposed methodology, authors should explain why they selected and combined Flesch Reading Ease with Flesch Kincaid and Gunning Fog Index Formula. How these 3 methods were integrated?
The three formulas are used to get the best readability score for new text, these formulas are used as separated not combined. We check new text in each of them by using the algorithm for each of them to check readability before and after to proof that the text readability has been improved.
4. Authors did not explain the role and impact of weighting factors used in formulas (line 353, 361) that were integrated in software , but they have include formulas with weighting factors as they are introduced by original authors. It would be very useful for readers to know the reasons for all weighting factors (206.835, 84.6...0,39, 11.8, 15.9, 0.4). It is supposed that these data about the roots for these weight factors could be found in the source work from original authors of these formulas. Authors should provide several sentences to clarify why to use these weigh factors in these formulas, according to original authors of formulas...
We have tried our best to find out explanation to these factor, however, we donot find. IWe will be more than happy and grateful if you know a reference which we could use to explain how and why these factors have been used.
5. Authors have only one Methodology title having contents that should be divided in separate sections. Separating sections will make the text more readable.
6. Instead of Methodology, there should be titie: "Proposed methodology of code comments evaluation" with subsections: The proposed approach (including figure 2, the algorithm of the approach with figure 3, the use of formulas lines 350-370), Developed software tool (line 319 - 339, 343 -349) An example of system use (From line 371-423)
Methodology has been changes to the proposed title and splitted into the suggested subsections.
7. Before result section there should have title "Research methodology" with explanation of methods, tools, sample for both: testing and survey. It is especially important to make clear explanation of data collection related to survey, since there is also GitHub projects data collection mentioned in line 463, so it is not clear enough how the data collection from Github is performed (access to source code, code selection...sample formation) and how the Github data collection is related to survey. Finally, the research methods should briefly mention and explain also the methods for data analysis (demonstrated with results starting from 539...).
explanation has been added to the proposed methodology section lines 341-347
8. Result section should have title "Results" and subsections: Results of software testing, Results of conducted survey (with another subsections - Results of survey with programmers, Results of survey with students).
Results section has been used and divided into subsections as suggested by the reviewer.
9. Figure 15 is too small and therefore unreadable - percentage in pies could not be read ...!!!
Figure 15 has been changed and made clearer and readable.
10. Conclusion section includes details about the research methodology (data collection, relation with Gihthub, evaluation from the instructor...) All these details should be provided before results section, in research methodology section. Conclusion should be more concise and to make summary of results and provide motivation with shorter discussion about the advantages and disadvantages of the proposed approach.
we have added the details of data collection to the proposed methodology.
for more information please see the attached file with track changes.

Reviewer 3 Report
The edited version of this manuscript is gained in quality; however, it may require additional work to make it perfect.
The introduction could be expanded to better introduce the reader toward the studied topic.
The introduction does not show a deep preview of the conducted study, I would suggest following a standard/common presentation style.
Literature Review section is quite small to be called literature review, I would suggest to call it background.
Author Response
The introduction could be expanded to better introduce the reader toward the studied topic.
The introduction does not show a deep preview of the conducted study, I would suggest following a standard/common presentation style.
Literature Review section is quite small to be called literature review, I would suggest to call it background.
The introduction has been improved according to your comments. this is shown from line 46 to line 60.
